# The Distribution Characteristics of Aerosol Bacteria in Different Types of Pig Houses

**DOI:** 10.3390/ani12121540

**Published:** 2022-06-14

**Authors:** Huan Cui, Cheng Zhang, Juxiang Liu, Shishan Dong, Kui Zhao, Ligong Chen, Zhaoliang Chen, Yucheng Sun, Zhendong Guo

**Affiliations:** 1Changchun Veterinary Research Institute, Chinese Academy of Agriculture Sciences, Changchun 130122, China; cuihuan1349@163.com (H.C.); zhangcheng1349@163.com (C.Z.); yuchengsunw@163.com (Y.S.); 2College of Animal Medicine, Jilin University, Changchun 130062, China; kuizhao@jlu.edu.cn; 3College of Veterinary Medicine, Hebei Agricultural University, Baoding 071000, China; ljx0315@126.com (J.L.); dongshishan@163.com (S.D.); clg01@163.com (L.C.); zl981429835@163.com (Z.C.)

**Keywords:** bacterial aerosol, closed pig house, growth stage, 16S rRNA gene sequencing

## Abstract

**Simple Summary:**

Microbial aerosols from pig houses can be released into the environment, posing a serious threat to biosafety and public health. At present, there are few studies on the structural characteristics of aerosol bacteria in piggeries at different growth stages. It is important to understand the characteristics of aerosol bacteria in pig houses to solve the problems of air pollution and disease control in pig houses at different growth stages. In this study, bacterial aerosol concentrations and bacterial communities were compared in pig houses at different growth stages in Hebei Province, China. It was found that bacterial concentrations, community richness, and diversity in the air increased with the age of pigs. There are many pathogenic bacteria in the microbial aerosols of piggery. Our study highlights the importance of more comprehensive research and analysis of microbial aerosols in pig houses. Precautions for air pollution should be instituted in pig houses, including wearing masks, rigorous disinfection, and hygiene procedures.

**Abstract:**

With the development of modern pig raising technology, the increasing density of animals in pig houses leads to the accumulation of microbial aerosols in pig houses. It is an important prerequisite to grasp the characteristics of bacteria in aerosols in different pig houses to solve the problems of air pollution and disease prevention and control in different pig houses. This work investigated the effects of growth stages on bacterial aerosol concentrations and bacterial communities in pig houses. Three traditional types of closed pig houses were studied: farrowing (FAR) houses, weaning (WEA) houses, and fattening (FAT) houses. The Andersen six-stage sampler and high-volume air sampler were used to assess the concentrations and size distribution of airborne bacteria, and 16S rRNA gene sequencing was used to identify the bacterial communities. We found that the airborne bacterial concentration, community richness, and diversity index increased with pig age. We found that *Acinetobacter*, *Erysipelothrix*, *Streptococcus*, *Moraxella*, and *Aerococcus* in the microbial aerosols of pig houses have the potential risk of causing disease. These differences lead us to believe that disinfection strategies for pig houses should involve a situational focus on environmental aerosol composition on a case-by-case basis.

## 1. Introduction

With the development of modern pig raising technology, the increasing density of animals in pig houses has resulted in poor air quality in pig houses [1]. The accumulation of microbial aerosols in pig houses poses a serious public health threat [1,2,3]. Microbial aerosols play an important role in the transmission of pathogens [4]. Previous studies have shown that exposure to high concentrations of microbial aerosols is associated with some diseases, such as cardiovascular disease [5,6], atherosclerosis [7], skin lesions [8], and respiratory diseases [9,10]. Pig farmers are at a higher risk of developing respiratory symptoms than those in office or normal human living environments [11,12,13].

Previous studies have shown that the pig gut microbiome varies significantly at different growth stages, while the air microbiota in pig houses is mainly derived from feces [14,15]. However, in different growth stages, how the aerosol bacteria in pig houses change, which of these bacteria are pathogenic bacteria, and the particle size distribution have not been identified. Therefore, it is necessary to study the structure and particle size distribution of microorganisms in pigsties at different stages. Moreover, previous studies have shown that bacteria play a dominant role in microbial aerosols in pig houses [16], and the most dominant phyla in pig houses are Firmicutes and Proteobacteria [17,18]. Potential pathogenic bacteria (such as *Acinetobacter*, *Erysipelothrix*, *Streptococcus*, etc.) in microbial aerosols of pig house bacteria have been widely studied by researchers [19,20,21,22]. Because of technical limitations, culture-based methods are not able to comprehensively analyze the diversity of airborne microorganisms in pig houses. The method of 16S ribosomal RNA (rRNA) gene sequence analysis has been widely used to describe the diversity of bacterial communities [23].

Pigs are an important group of commercial species raised in Hebei Province, China. However, because of the lack of related large-scale pig breeding enterprises, most pig farms are still based on traditional closed breeding, and the structure of the pig house is relatively simple. Furthermore, there is no automatic excrement cleaning or automatic control of the environment in pig house-related equipment. During the winter, there is an increase in respiratory illness among farmworkers, as microbial aerosol concentrations tend to increase with particulate concentrations because of reduced ventilation in pig houses [24,25].

In this study, the concentrations of aerosol bacteria and bacterial communities, as well as bacterial community diversity and community composition of three types of pig houses (FAR, WEA, and FAT), were compared in three typical closed pig farms in Hebei Province. Our data provide a novel basis for the threat to public health of pig house aerosol bacteria.

## 2. Materials and Methods

### 2.1. Characteristics of Research Objects

During the winter of 2019, samples were taken from three representative pig farms in Hebei Province, China, all of which belonged to the same swine industry corporation. Pig farms were anonymized according to the company’s requirements. Each pig farm had four types of pig houses: a gestating stable, a farrowing (FAR) stable, a weaning (WEA) stable, and a fattening (FAT) stable. According to the company’s requirements for the safety of pregnant sows, we did not sample gestation houses. The FAR stables consisted of post-farrowing sows and preweaning pig houses, the WEA stables raised postweaning pigs, and the FAT stables raised fattening pigs. For safe production, the FAR stables were positioned in a separate area, while the WEA and FAT stables were located in another. Nursery pigs could be transported from FAR stables to WEA stables through special transport channels.

For the three types of pig houses (breed: Danish landrace), the stocking density was 3.7–4.2 m^2^/head for parturient sows, 0.5–0.8 m^2^/head for nursery pigs, and 1–1.5 m^2^/head for finishing pigs. Indoor structures of pig houses included sheet steel roofs and brick walls using thermal insulation material, with ventilation and a water curtain to maintain a relatively constant temperature and humidity. Each pig house occupied an area of approximately 60 × 15 m^2^ and was equipped with a double-row structure. Every 5 m, there was a corral in the pig house, and each corral was equipped with a deep pit with slats to collect pig manure, which was cleaned every 15 days. There were no disease outbreaks on any of the farms during our investigation, and samples were collected ten days after fecal cleaning.

### 2.2. Air Sampling and Cultivation of Isolated Bacteria

An Andersen six-stage sampler (TE-20-800, TISCH, Cleves, OH, USA) was used to identify the size distribution characteristics of culturable bacteria in airborne microorganisms. For each pig house, one sampling point was set in the middle of the passage of the pig house with a height of 0.5 m (the breathing height of pigs) above the ground. The sampling rate was 28.3 L/min for 3 min [2]. The sampler was routinely sterilized with 75% ethanol and dried before sampling. Luria-Bertani (LB) agar medium (Solarbio Science & Technology Co., Ltd., Beijing, China) was used to collect culturable bacteria. The culture plates were then incubated at 37 °C for 24 h. The particle size ranges of the air sampler were 0.65–1.1 μm, 1.1–2.1 μm, 2.1–3.3 μm, 3.3–4.7 μm, 4.7–7.0 μm, and ≥7.0 μm for stages 6, 5, 4, 3, 2, and 1, respectively. Each sample was evaluated in triplicate, and bacterial colonies were counted and calculated according to a previous method to obtain the number of bacterial colonies per cubic meter of air [26].

### 2.3. Collection of Microbial Aerosols

The method for the collection of microbial aerosols was performed according to our previous study [23]. In brief, a high-volume air sampler (HH02-LS120 of Beijing Huaruihean Technology Co., Ltd., Beijing, China) with 20.32 × 25.4 cm^2^ Tissuquartz filters (PALL, Port Washington, NY, USA) was used to collect microbial aerosols in the pig house. The sampler was placed in the middle passage of the pig house at a height of 0.5 m above the ground, with a sampling efficiency of 1000 L/min and a sampling time of 12 h [27]. A total of 27 samples were collected from three farms (9 samples per farm), and 3 replications were collected from each pig house on the same farm and mixed together for analysis. Each membrane was cut into an average of 8 pieces with sterile surgical scissors, with a weight error of ±1 mg for each part. Ultrapure water (ST876, Yaji, Shanghai, China) was added to elute the particles. Homogenization was carried out with a homogenizer, and centrifugation was performed at 25,000× *g* for 10 min at 4 °C. All the filters were sterilized by baking in a muffle furnace at 500 °C for 48 h before sampling. The filters were stored in a mobile refrigerator and transported to the laboratory after collection. Samples were stored at −80 °C until analysis.

### 2.4. 16S rRNA Gene Sequencing

The total genome DNA extraction method and PCR conditions of microbial aerosol samples were performed as described in previous studies [27,28,29]. A NanoDrop 2000 spectrophotometer (Thermo Fisher Scientific, Wilmington, DE, USA) was used to detect the concentration and quality of extracted DNA (OD260/OD280). PCR amplification was performed after qualified detection. The unqualified samples were extracted again until they passed the test. For bacterial diversity analysis, primers 515F (5′-GTGCCAGCMGCCGCGGTAA-3′) and 806R (5′-GGACTACHVGGGTWTCTAAT-3′) were used to amplify the V3/V4 region of the 16S rRNA gene [30]. For 16S rRNA high-throughput sequencing, the PCR products from each sample were transferred to Beijing Novogene Bioinformatics Technology Co., Ltd. (Novogene, Beijing, China). All original sequences were deposited in GenBank under accession number PRJNA815738.

### 2.5. Statistical Analysis

QIIME software was employed to remove the poor-quality sequences and obtain high-quality data to control the sequence quality. OTUs (operational taxonomic units) were clustered using UPARSE software (v7.0.1001), which had 97% identity for 16S rRNA sequences [31]. Species annotation was performed on the OTU sequences, and the Mothur method and the SSUrRNA database [32] of SILVA132 were used for species annotation analysis to obtain taxonomic information at each taxonomic level (phylum and genus) and for community composition analysis. The observed-species, Shannon, and Chao1 indices in each sample were calculated with QIIME (version 2.0). Nonmetric multidimensional scaling (NMDS) based on an unweighted Unifrac distance matrix was used to estimate beta diversity. Differences in microbial community structure between groups were also analyzed by principal component analysis (PCA) [33,34]. All data were subjected to one-way analysis of variance (one-way ANOVA) using GraphPad Prism 8 software to determine statistically significant differences (*p* < 0.05). All analyses were conducted in triplicate and represented at least three separate experiments. The results are presented as means ± standard deviation (SD).

## 3. Results

### 3.1. Size Distribution of Culturable Airborne Bacteria

The concentrations of culturable bacteria in the air of the three pig house types (FAR, WEA, and FAT) are shown in Figure 1. The concentrations of airborne bacteria in the FAR, WEA, and FAT stables were 8.092 ± 1.089 × 10^3^ CFU/m^3^, 15.669 ± 1.899 × 10^3^ CFU/m^3^, and 35.027 ± 2.859 × 10^3^ CFU/m^3^, respectively. Bacterial aerosol concentrations were significantly higher in the WEA and FAT stables than in the FAR stables (*p* < 0.001) and significantly higher in the FAT stables than in the WEA stables (*p* < 0.001).

Figure 2 shows the size distribution of culturable airborne bacteria in the three types of pig houses (FAR, WEA, and FAT). In the FAR stables, bacteria with a particle size greater than 3.3 µm (Andersen six-stage sampler, stages 1–3) accounted for 68.2%, while bacteria with a particle size range of 0.6 to 3.3 µm (Andersen six-stage sampler, stages 4–6) accounted for 31.8%. In the WEA stables, a total of 56.2% of airborne bacteria had a particle size greater than 3.3 µm, and 43.8% had a particle size range of 0.6 to 3.3 µm. The majority of bacteria in the FAT stables were found in stage 1 (≥7.0 μm) (28.2%), and stage 5 (1.1–2.1 μm) (7.5%) was the least common.

### 3.2. Sequencing Analysis

Nine air samples from three pig house types (FAR, WEA, and FAT) were sequenced for the bacterial V3/V4 region. After filtering out the poor-quality sequences, 85,524 effective reads were obtained. The sequences were clustered into 3362 operational taxonomic units (OTUs) with 97% identity, among which the OUTs of the FAR stables were 1.026 ± 0.215 × 10^3^ and the OUTs of the WEA stables were 1.405 ± 0.219 × 10^3^. In the FAT stables, the OUTs were 1.703 ± 0.295 × 10^3^ (Figure 3A). Compared with the FAR stables, the WEA stables showed a significantly increased number of OUTs (*p* < 0.01). The number of OUTs in the FAT stables was significantly higher than that in the FAR and the WEA stables (*p* < 0.05). At the same time, the dilution curves of all samples were infinitely close to saturation, indicating that most bacteria in the samples were detected, ensuring the reliability and stability of subsequent analysis.

### 3.3. Diversity of Airborne Bacterial Communities

The α-diversity and β-diversity of bacterial communities in the FAR, WEA, and FAT stables are shown in Figure 3B,C and Figure 4. The Chao1 index and Shannon index were evaluated in the α-diversity analysis (Figure 3B,C). Compared with the WEA and FAR stables, the Chao1 index and Shannon index of the FAT stables were significantly higher (*p* < 0.001) (Figure 3B,C). The Chao1 index and Shannon index of the WEA stables were significantly higher than those of the FAR stables (*p* < 0.05) (Figure 3B,C). Figure 4 shows the principal component analysis (PCA) of β-diversity. Principal component analysis (PCA) extracted the two axes that best reflected the differences between samples, and each point represented a sample. The closer these points were on the PCA plot, the more similar the community composition of the samples was. The PCA results showed (Figure 4) that there were certain differences in the bacterial communities of the FAR, WEA, and FAT stables. The multiple response permutation programs (MRPP) test showed significant differences among the three types of pig houses (*p* < 0.05).

### 3.4. Composition of the Airborne Bacterial Community

At the phylum level, Firmicutes and Proteobacteria were the most abundant in the bacterial community composition, accounting for 68.86% and 20.23% in the FAR stables, 66.94% and 3.54% in the WEA stables, and 74.95% and 5.4% in the FAT stables, respectively. Figure 5 and Appendix A show the 10 dominant genera in the samples. Our data show that the dominant bacterial genera in the FAR samples were *Clostridiales* (21.70%), *Psychrobacter* (16.07%), *Streptococcus* (6.78%), *Terrisporobacter* (4.96%), *Lactobacillus* (4.47%), *Corynebacteriaceae* (2.71%), *Aerococcus* (2.03%), *Cyanobacteria* (1.88%), *Staphylococcus* (0.58%), and *Prevotellaceae* (0.06%). The dominant bacterial genera in the WEA samples were *Lactobacillus* (16.24%), *Streptococcus* (12.78%), *Clostridiales* (5.48%), *Prevotellaceae* (4.73%), *Corynebacteriaceae* (4.14%), *Cyanobacteria* (2.54%), *Aerococcus* (1.70%), *Terrisporobacter* (1.60%), *Staphylococcus* (0.72%), and *Psychrobacter* (0.08%). The dominant bacterial genera in the FAT samples were *Clostridiales* (29.44%), *Corynebacteriaceae* (9.97%), *Streptococcus* (9.56%), *Aerococcus* (5.49%), *Terrisporobacter* (3.40%), *Lactobacillus* (2.12%), *Psychrobacter* (1.95%), *Cyanobacteria* (1.87%), *Streptococcus* (1.74%), and *Prevotellaceae* (0.12%). Heatmaps for the top 35 genera of all samples are shown in Figure 6. According to the Human Pathogenic Bacteria Directory provided by the Ministry of Health of China (MOHC), the potentially pathogenic bacteria in the FAR stables were *Acinetobacter*, *Streptococcus*, and *Erysipelothrix*; the potentially pathogenic bacteria in the WEA stables were *Streptococcus*.

## 4. Discussion

In this study, three types of pig houses (FAR, WEA, and FAT) of three typical closed pig farms in Hebei Province were investigated. The aerosol bacterial concentration and bacterial community, as well as the bacterial community diversity and composition of the three types of pig houses, were compared to provide a data basis and reference for healthy pig breeding and the threat of microbial aerosols to public health in pig houses.

Previous studies have shown that bacteria dominate microbial aerosols in pig houses [16], so this study focused on bacterial microbial aerosols. Our study shows that airborne bacterial concentrations in pig houses increase as pigs age. The concentrations of culturable airborne bacteria were the lowest in FAR and significantly higher in WEA and FAT (*p* < 0.001). This may be due to the increased activity and excretion of pigs as they grow and increased feeding density, resulting in increased concentrations of bacterial aerosols. The numbers of culturable bacteria measured in the three types of stables (FAR, WEA, and FAT) were 8.092 ± 1.089 × 10^3^ CFU/m^3^, 15.669 ± 1.899 × 10^3^ CFU/m^3^, and 35.027 ± 2.859 × 10^3^ CFU/m^3^, respectively. According to studies, the culturable bacterial concentration in the WEA stables of Jiangsu, China, was 3.460× 10^3^ CFU/m^3^ (1), and the culturable bacterial concentrations in the air of three types of pig houses (FAR, WEA, and FAT) in Korea were 2.322 ± 1.352 × 10^3^ CFU/m^3^, 6.124 ± 1.527 × 10^3^ CFU/m^3^, and 12.470 ± 4.869 × 10^3^ CFU/m^3^ [35]. In our investigation, the concentration of bacterial microbial aerosols was greater than that in previous studies. The pigs, the construction of the barns, the methods of management, and the indoor climate may all have a role in the fluctuation in microbiological concentrations of bacterial aerosols [36].

Different particle sizes of bacterial aerosols enter the respiratory system in different locations. Particles of 4.7–7 µm in diameter can enter the nasal cavity and upper respiratory tract; particles of 2.1–4.7 µm can be deposited in the bronchi, causing asthma and other diseases; however, those less than 2 µm in diameter can penetrate the alveoli and enter the respiratory tract, causing alveolar inflammation and other diseases. The smallest fraction of the fine particles may have a disproportionately large role. Although fine particles are only a small proportion, they still pose a serious threat to the respiratory health of pigs. In the FAT stables, 71.9% of airborne bacteria had a particle size larger than 3.3 µm, and 28.1% had a particle size ranging from 0.6 to 3.3 µm. Aerosols with a particle size of less than 3.3 µm are defined as fine particles, and those larger than 3.3 µm are defined as coarse particles. Culturable bacteria in the air predominate in coarse particles, but not in fine particles, similar to our results [2]. Previous research showed that coarse particles accounted for 64% and fine particles accounted for 36% of particles in the FAR stables. In the WEA stables, coarse particles accounted for 60%, and fine particles accounted for 40% of the total particles. The proportions of coarse particles and fine particles were 63% and 37%, respectively [35]. The smallest fraction of the fine particles may have a disproportionately large role. Although fine particles are only a small proportion, they still pose a serious threat to the respiratory health of pigs. The results were similar to those in the FAR and WEA stables, but the proportion of coarse particles in the FAT stables was higher, which may have been caused by various factors, such as pig breed, stocking density, climate, and the structure of the barns.

We found that the bacterial community α-diversity index of microbial aerosols increased gradually with the age of pigs. The Chao1 index and Shannon index of the FAT stables were significantly higher than those of the FAR and WEA stables (*p* < 0.001). PCA showed that the composition of the bacterial community also changed significantly (*p* < 0.05). As animals grow, their activity and excretion increase and their metabolic levels change, which could cause remarkable changes in the microbial community structure in the air of feedlots [37]. Firmicutes and Proteobacteria were the dominant phyla in the three types of pig houses, which is consistent with previous studies [17,18,20]. Firmicutes are the most abundant phylum in the intestinal microbiome of pigs at all growth stages [38], suggesting that the dominant bacteria in the air of the pig house may originate from feces [39]. Proteobacteria is the largest phylum of bacteria, including many pathogenic bacteria, such as Escherichia coli, Salmonella, Vibrio cholera, Helicobacter pylori, and other well-known species [40,41,42,43], while some members of Proteobacteria play an important ecological role in various environments [44].

According to the human pathogenic bacteria directory provided by the Ministry of Health of China (MOHC), the potentially pathogenic bacteria in the FAR stables were *Acinetobacter*, *Streptococcus*, and *Erysipelothrix*; those in the WEA stables were *Streptococcus*. *Acinetobacter* widely exists in the environment, and studies have shown that the mortality rate of patients with underlying diseases infected with *Acinetobacter* is much higher than that of other patients [45,46,47]. Meanwhile, the number of *Acinetobacter* strains resistant to multiple antibiotics has increased significantly in recent years, suggesting potential biosafety risks [48]. *Erysipelothrix* is extensively distributed in the environment and may survive for a long period [49,50]. *Erysipelothrix* is a pathogen associated with many occupational diseases, and its pathogenesis may be related to hyaluronidase and neuraminidase produced by bacteria [19]. *Streptococcus* suis, an important zoonotic pathogen in the genus *Streptococcus*, lives naturally in the respiratory system of pigs, particularly the upper respiratory tract, and is found in nearly every country [22]. Previous studies have reported that *Streptococcus* has been found in nasal swabs of pig workers [51]. Thus, the prevention and control of *Streptococcus* deserve wider attention. *Moraxella* in WEA and *Aerococcus* in FAT, in addition to the list of human pathogenic bacteria, have been identified as potentially pathogenic concerns that should be monitored. *Moraxella* catarrhalis is a pathogenic strain of *Moraxella* that has been found in the human respiratory system [52]. In the respiratory system, *Moraxella lacunata* and *Moraxella nonliquefaciens* are also opportunistic [53]. *Aerococcus* has been recovered from clinical samples of pigs in a growing number of investigations, indicating that it is pathogenic [39,54,55]. Previous research has found that microbial aerosols in pig houses include many *Aerococcus*, and pig employees working in an environment with many *Aerococcus* are at risk of becoming sick [1]. The source of the pathogenic bacteria of this genus is unclear, but pathogenic bacteria are likely to spread through pig houses to the outside world, and pig employees are likely to carry these infections on their clothes or in their lungs, resulting in a wide range of biological safety hazards and concerns [2,46,56].

## 5. Conclusions

This study showed that with the increase in weekly pig age, the concentration of airborne bacteria, the species, and the diversity index of the bacterial community in pig houses has also increased gradually. The pathogenic bacteria Acinetobacter, Erysipelothrix, Streptococcus, Moraxella, and Aerococcus were detected in three types of pig houses. These bacteria with potential risk to pig and human health should be monitored. The number of disinfections should be increased to 2–3 times a week or every other day. A disinfectant should be replaced after 2–3 weeks of continuous use for another disinfectant to avoid pathogenic microorganisms producing drug resistance and reduce the disinfection effect. Spray disinfection should be used to kill airborne microorganisms in piggeries to make the piggeries cleaner. These measures may help optimize the piggery environment. From the point of view of personnel, farmers should wear masks, hats, and overalls to combat the effects of air pollution when working on farms. In this study, we emphasize the necessity of long-term detection of microbial aerosols in pig houses and the importance of more comprehensive research and analysis of microbial aerosols in livestock and poultry houses.

## Figures and Tables

**Figure 1 animals-12-01540-f001:**
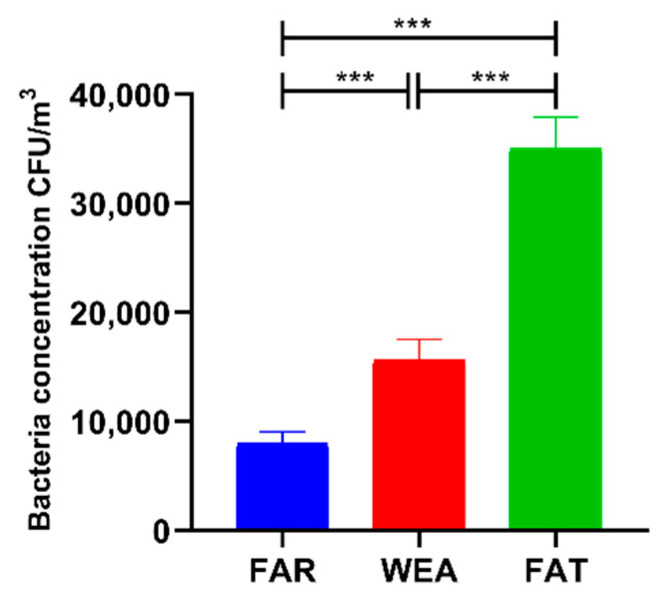
Concentrations of culturable airborne bacteria in different types of pig houses using an Andersen sampler (data presented as means ± SD. *** *p* < 0.001). FAR, farrowing houses; WEA, weaning houses; FAT, fattening houses.

**Figure 2 animals-12-01540-f002:**
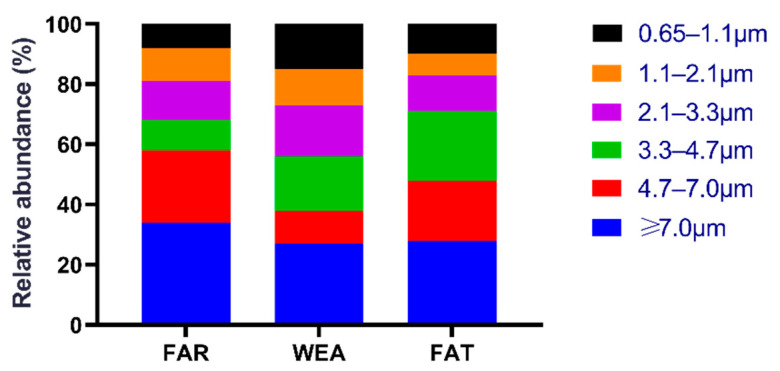
The distribution ratio of culturable airborne bacteria in different types of pig houses. FAR, farrowing houses; WEA, weaning houses; FAT, fattening houses.

**Figure 3 animals-12-01540-f003:**
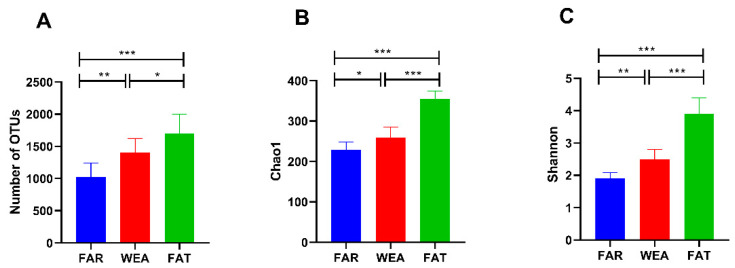
Alpha diversity of the bacterial community in different types of pig houses. (**A**) Observed species; (**B**) Chao1 index; (**C**) Shannon index (data presented as the means ± SDs. * *p* < 0.05, ** *p* < 0.01, *** *p* < 0.001). FAR, farrowing houses; WEA, weaning houses; FAT, fattening houses.

**Figure 4 animals-12-01540-f004:**
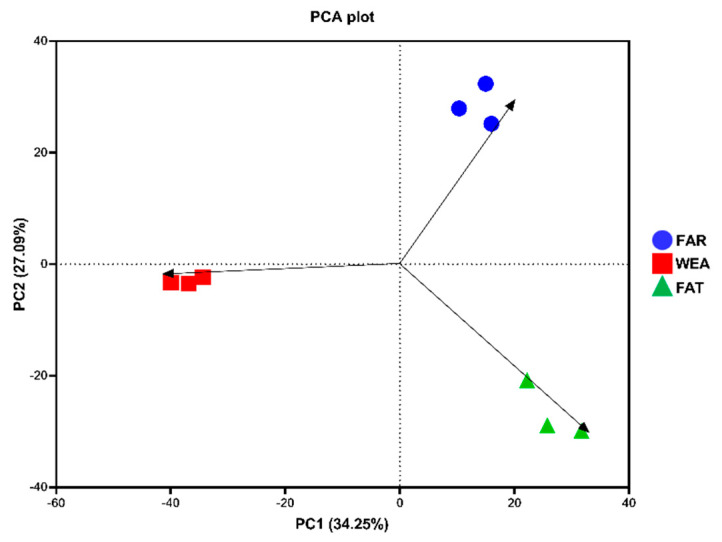
Principal component analysis (PCA) of samples from different types of pig houses. FAR, farrowing houses; WEA, weaning houses; FAT, fattening houses.

**Figure 5 animals-12-01540-f005:**
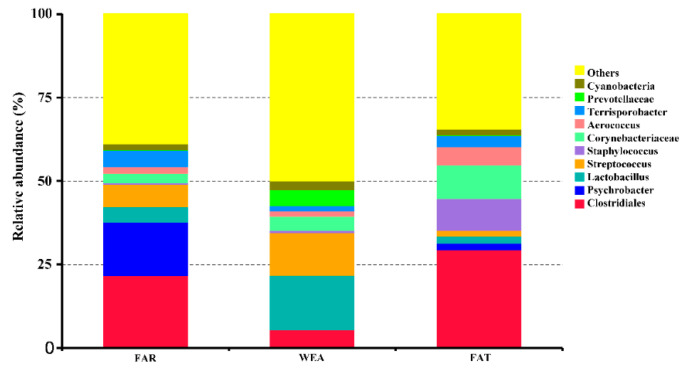
Relative abundances of the airborne bacterial genera (%).

**Figure 6 animals-12-01540-f006:**
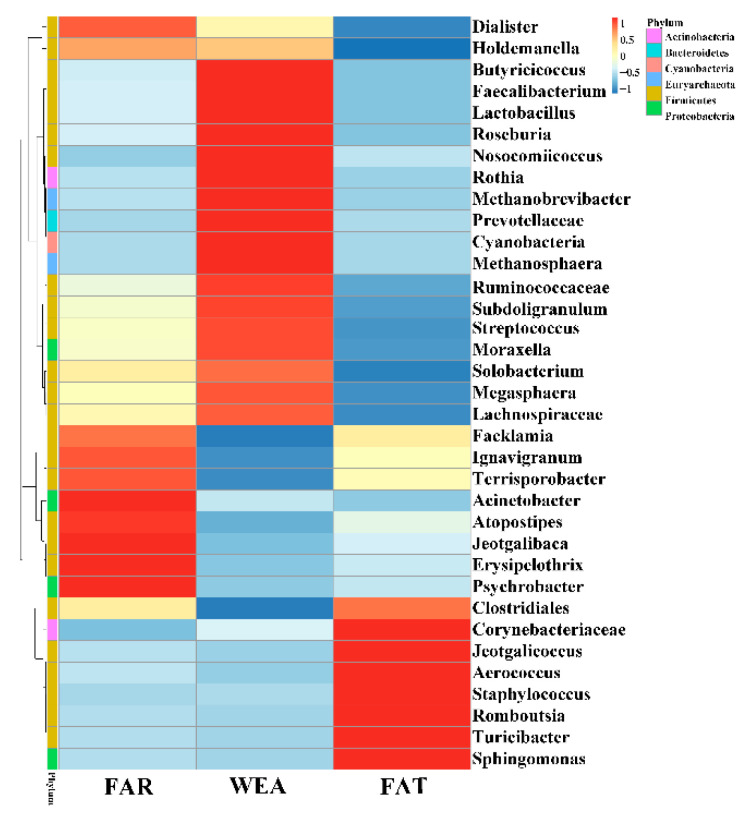
Heatmap of airborne bacterial relative abundance at the genus level. FAR, farrowing houses; WEA, weaning houses; FAT, fattening houses.

## Data Availability

The study’s original contributions are included in the article; further inquiries can be directed to the corresponding authors.

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
