# Peer review of "The Distribution Characteristics of Aerosol Bacteria in Different Types of Pig Houses"

_animals, 2022, doi:10.3390/ani12121540_

Round 1

Reviewer 1 Report

I have posted the comments to the article below

Simple Summary: the Authors write that it is important to understand the characteristics of airborne bacteria in the pig house in order to solve the problem of air pollution and disease control. However, the manuscript does not provide a solution to this problem.

Abstract: it is very general. The abstract of the article always ends with outlining the benefits of the obtained research results and recommendations as a solution to the presented problem. I miss such information in the abstract. I think that the Authors should focus more on the innovation of their own research results.

Keywords: I propose to change the microbial aerosol to a bacterial aerosol, because the Authors only tested bacteria and therefore the last keyword (bacteria) should be removed.

The purpose of the research should be formulated in more detail.

Materials and Methods:

2.1. Sample Collection: at this point, the Authors characterized the air sampling sites, therefore the title of this chapter should be: Characteristics of research objects.

2.2. Determination of Culturable Airborne Bacteria: the title of this point is also inadequate to the content. The Authors describe the air sampling technique using the Andresen 6-stage cascade impactor and describe the conditions under which they incubated the isolated bacteria. The title should read: Air sampling and cultivation of isolated bacteria.

2.3. Collection of Microbial Aerosols: this point is incomprehensible to me. Why did the Authors use two air samplers? For comparison purposes? It is nowhere explained.

2.416. S rRNA Gene Sequencing, it should read: 2.4. 16S rRNA Gene Sequencing

Results

3.1. Size Distribution of Culturable Airborne Bacteria: in the microbiological nomenclature, the number of colony forming units in m3 of air is presented as follows: 8.092±1.089·103 CFU·m-3, 15.669 ±1.899·103 CFU·m-3 i 35.027±2.859·103 CFU·m-3 (line 153). I believe Authors should change this throughout the manuscript.

Figure 2 shows that in all research objects, the greatest number of bacteria was isolated in the first segment of the impactor, with the diameter of the inlet holes ≥7.0 µ, and the least in the sixth segment (hole diameter 0.65-1.1 µ). The Andersen impactor is designed to measure the concentration of specific size distributions of microorganisms in an aerosol (air in the tested environment). The structure of the impactor, the diameter of the holes of successive segments, together with the flow applied, allow the distribution of the aerosol fraction into groups with aerodynamic diameters corresponding to the penetration of various places in the respiratory system. The Authors did not describe how the obtained results may affect the respiratory system of pigs.

Discussion: has been properly conducted, but the information contained in the last paragraph (lines 270-301) is not the result of the research conducted by the authors. The Authors took air at a height of 0.5 m, i.e. in the respiratory zone of pigs. Did the Authors also check the distribution of bacteria in the respiratory zone of humans and performed antibiotic resistance tests of the isolated bacteria?

Conclusions: poorly worked out. The Authors write that the concentration of bacteria in the air, the species, and the diversity index increased with the age of the pigs. It is evident that as each mammal ages, the type of food changes, it resulting in changes in the gastrointestinal microbiome and the type of microorganisms that are excreted. Such results were to be expected and the Authors documented them in detail in the discussion. It is well known that bacteria are present in pig houses and other farm facilities, so what has this research brought about? Can the Authors propose a solution to this problem? Maybe the pig hauses should be cleaned more often than every 15 days or disinfected? It seems to me that this chapter should be redrafted.

Author Response

Response to Reviewer 1 Comments

  • Simple Summary: The Authors write that it is important to understand the characteristics of airborne bacteria in the pig house in order to solve the problem of air pollution and disease control. However, the manuscript does not provide a solution to this problem.

Response 1: We thank the reviewer for pointing this out. We added a solution to this problem: “Precautions for air pollution should be instituted in pig houses, including wearing masks and rigorous disinfection and hygiene procedures” (lines 21-22).

  • Abstract: it is very general. The abstract of the article always ends with outlining the benefits of the obtained research results and recommendations as a solution to the presented problem. I miss such information in the abstract. I think that the Authors should focus more on the innovation of their own research results.

Response 2: We agree with the comment and have revised the abstract to address your concerns and hope that it is now clearer (lines 23-36). The sentences in the revised manuscript are as follows:

With the development of modern pig raising technology, the increasing density of animals in pig houses leads to the accumulation of microbial aerosols in pig houses. It is an important prerequisite to grasp the characteristics of bacteria in aerosols in different pig houses to solve the problems of air pollution and disease prevention and control in different pig houses. The purpose of this study was to investigate the effects of growth stages on bacterial aerosol concentrations and bacterial communities in pig houses. Three traditional types of closed pig houses were studied: farrowing (FAR) houses, weaning (WEA) houses, and fattening (FAT) houses. The Andersen 6-stage sampler and high-volume air sampler were used to collect air samples, and 16S rRNA gene sequencing was used to identify the bacterial communities. We found that the airborne bacterial concentration, community richness, and diversity index increased with pig age. We found that Acinetobacter, Erysipelothrix, Streptococcus, Moraxella, and Aerococcus have the potential risk of disease in the microbial aerosols of pig houses. These differences lead us to believe that disinfection strategies for pig houses should involve a situational focus on environmental aerosol composition on a case-by-case basis.

  • Keywords: I propose to change the microbial aerosol to a bacterial aerosol, because the Authors only tested bacteria and therefore the last keyword (bacteria) should be removed.

Response 3: We agree with the comment and have changed “microbial aerosol” to “bacterial aerosol”, and the last keyword “bacteria” has been removed (line 37).

  • The purpose of the research should be formulated in more detail.

Response 4: We thank the reviewer for pointing this out. We have revised the part about the purpose of our research: It is an important prerequisite to grasp the characteristics of bacteria in aerosols in different pig houses to solve the problems of air pollution and disease prevention and control in different pig houses. This work investigated the effects of growth stages on bacterial aerosol concentrations and bacterial communities in pig houses (lines 24-28).

  • 1. Sample Collection: at this point, the Authors characterized the air sampling sites, therefore the title of this chapter should be: Characteristics of research objects.

Response 5: We agree and have modified the title of this chapter to “Characteristics of research objects” (line 76).

  • 2. Determination of Culturable Airborne Bacteria: the title of this point is also inadequate to the content. The Authors describe the air sampling technique using the Andresen 6-stage cascade impactor and describe the conditions under which they incubated the isolated bacteria. The title should read: Air sampling and cultivation of isolated bacteria.

Response 6: Based on the reviewer’s suggestion, we changed the title of “2.2 Determination of Culturable Airborne Bacteria” to “2.2. Air Sampling and Cultivation of Isolated Bacteria” (line 97).

  • 3. Collection of Microbial Aerosols: this point is incomprehensible to me. Why did the Authors use two air samplers? For comparison purposes? It is nowhere explained.

Response 7: We are grateful for the suggestion. Our previously published research follows this model. Andersen’s six-stage sampler was used to identify the size distribution characteristics of culturable bacteria with LB agar medium (lines 112-114). However, there are some nonculturable bacteria in the air. In that case, a high-volume air sampler was used to collect all the microbial aerosols with Tissuquartz filters.

  • 416. S rRNA Gene Sequencing, it should read: 2.4. 16S rRNA Gene Sequencing

Response 8: Thank you for your kind comments. We changed the title of “2.416. S rRNA Gene Sequencing” to “2.4. 16S rRNA Gene Sequencing” (line 125).

  • 1. Size Distribution of Culturable Airborne Bacteria: In the microbiological nomenclature, the number of colony forming units in m3 of air is presented as follows: 8.092±1.089·103 CFU·m-3, 15.669 ±1.899·103 CFU·m-3, 35.027±2.859·103 CFU·m-3 (line 153). I believe Authors should change this throughout the manuscript.

Response 9: We thank the reviewer for this suggestion. As suggested by the reviewer, we have changed all the numbers to scientific notation as follows: 8.092±1.089·103 CFU/m3, 15.669±1.899·103 CFU/m3, 35.027±2.859·103 CFU/m3 (lines 157-158). 1.026±0.215·103, 1.405±0.219·103, 1.703±0.295·103 (lines 179-180) 8.092±1.089·103 CFU/m3, 15.669±1.899·103 CFU/m3, and 35.027±2.859·103 CFU/m3, 3.460·103 CFU/m3, 2.322±1.352·103 CFU/m3, 6.124±1.527·103 CFU/m3, 12.470±4.869·103 CFU/m3 (lines 242-246)

  • Figure 2 shows that in all research objects, the greatest number of bacteria was isolated in the first segment of the impactor, with the diameter of the inlet holes ≥7.0 µ, and the least in the sixth segment (hole diameter 0.65-1.1 µ). The Andersen impactor is designed to measure the concentration of specific size distributions of microorganisms in an aerosol (air in the tested environment). The structure of the impactor, the diameter of the holes of successive segments, together with the flow applied, allow the distribution of the aerosol fraction into groups with aerodynamic diameters corresponding to the penetration of various places in the respiratory system. The Authors did not describe how the obtained results may affect the respiratory system of pigs.

Response 10: We agree with this suggestion and have added the following description:

Different particle sizes of bacterial aerosols enter the respiratory system in different locations. Particles of 4.7-7 µm in diameter can enter the nasal cavity and upper respiratory tract; particles of 2.1-4.7 µm can be deposited in the bronchi, causing asthma and other diseases; however, those less than 2 µm in diameter can penetrate the alveoli and enter the respiratory tract, causing alveolar inflammation and other diseases. The smallest fraction of the fine particles may have a disproportionately large role. Although fine particles are only a small proportion, they still pose a serious threat to the respiratory health of pigs (lines 251-258).

  • Discussion: has been properly conducted, but the information contained in the last paragraph (lines 270-301) is not the result of the research conducted by the authors. The authors took air at a height of 0.5 m, i.e., in the respiratory zone of pigs. Did the Authors also check the distribution of bacteria in the respiratory zone of humans and performed antibiotic resistance tests of the isolated bacteria?

Response 11: Thank you for your comments. The purpose of our discussion in this section is to further analyze our findings to analyze the potential pathogenic genera and suggest potential risks. Numerous previous studies have followed the same pattern (Chen et al. 2021, Tang et al. 2021, Yan et al. 2019). You have provided extraordinary suggestions for our study, and in future studies, we will place more emphasis on examining the distribution of bacteria in the human respiratory zone and have tested the isolated bacteria for antibiotic resistance.

  • Conclusions: poorly worked out. The Authors write that the concentration of bacteria in the air, the species, and the diversity index increased with the age of the pigs. It is evident that as each mammal ages, the type of food changes, resulting in changes in the gastrointestinal microbiome and the type of microorganisms that are excreted. Such results were to be expected and the Authors documented them in detail in the discussion. It is well known that bacteria are present in pig houses and other farm facilities, so what has this research brought about? Can the Authors propose a solution to this problem? Maybe pig houses should be cleaned more often than every 15 days or disinfected? It seems to me that this chapter should be redrafted.

Response 12: We thank the reviewer for pointing out this issue. We indeed should propose a solution to this problem. As suggested by the reviewer, we added some suggestions for disinfection of piggery and personnel protection as follows: The number of disinfections should be increased to 2-3 times a week or every other day. A disinfectant should be replaced after 2-3 weeks of continuous use of another disinfectant to avoid pathogenic microorganisms producing drug resistance and reduce the disinfection effect. Spray disinfection should be used to kill airborne microorganisms in piggery to make the piggery cleaner. These measures may help to optimize the piggery environment. From the point of view of personnel, farmers should wear masks, hats and overalls to combat the effects of air pollution when working on farms (lines 317-324).

Reviewer 2 Report

Dear authors of the manuscript,

The work considering the aerosolisation of microbes from the pig farms is very actual and interesting for readers, since it is expected that many zoonozes can be transfered by the aerosolisation. The results are clear and address the general question about the community structure in the air, the viability by cultivation as well as distribution of microbes within the particular particle fractions. 

The main comments and improvement is for my opinion needed to be implemented is within the Discussion section. Many of the results here are related to the pathogenicity based on the related pathogenic species. Here it is needed to be careful and the main comment considering this problem are included in the attached annotated pdf file. Sorry for some wordings and strange colours, I had some problems transferring annotated file to the computer.

Author Response

Response to Reviewer 2 Comments

  • Line 117 here you missing detail how you proceed after sampling to the incubation.

Response 1: We agree with this suggestion and have added the postsampling process in the corresponding section as follows:

Each membrane was cut into an average of 8 pieces with sterile surgical scissors, with a weight error of ±1 mg for each part. Ultrapure water (ST876, Yaji, Shanghai, China) was added to elute particles. Homogenization was carried out with a homogenizer, and centrifugation was performed at 25000 × g for 10 minutes at 4°C (lines 118-122).

  • Line 121 change “16. S rRNA” to “16S rRNA”

Response 2: Thank you for your careful check. We changed the title of “2.416. S rRNA Gene Sequencing” to “2.4. 16S rRNA Gene Sequencing” (line 125).

  • Line 130 It is not known did you use DNA directly or PCR amplified. Please correct.

Response 3: Thank you for your comment. We changed “the extracted DNA samples” to “PCR products from each sample” (line 134).

  • Line 151-166 please shorten. You are repeating. The same result is written the same for all samples.

Response 4: We thank the reviewer for pointing out this issue and have modified the text about particle size distribution as follows:

Figure 2 shows the size distribution of culturable airborne bacteria in the three types of pig houses (FAR, WEA, and FAT). In the FAR, bacteria with a particle size greater than 3.3 µm (Andersen 6-stage sampler stage 1-3) accounted for 68.2%, while bacteria with a particle size range of 0.6 to 3.3 µm (Andersen 6-stage sampler stage 4-6) accounted for 31.8%. In the WEA, a total of 56.2% of airborne bacteria had a particle size greater than 3.3 µm, and 43.8% had a particle size range of 0.6 to 3.3 µm. The majority of bacteria in the FAT were found in stage 1 (≥ 7.0 μm) (28.2%), and stage 5 (1.1-2.1 μm) (7.5%) was the least common (lines 161-168).

  • Line 202 please add vector to understand the meaning.

Response 5: Thank you for your comment. We have followed your suggestion, and we have added vectors to Figure 4.

  • Line 203 this is not clear. Put in a table.

Response 6: Thank you for your comment. We have added this form to the supplemental materials as you suggested.

  • Line 210 you are mentioning only pathogenic bacteria what about beneficial?

Response 7: Thank you for your suggestions. We have revised the description of the manuscript. We have changed the description as you suggested, and the new description is as follows:

Our data show that the dominant bacterial genera in the FAR samples were Clostridiales (21.70%), Psychrobacter (16.07%), Streptococcus (6.78%), Terrisporobacter (4.96%), Lactobacillus (4.47%), Corynebacteriaceae (2.71%), Aerococcus (2.03%), Cyanobacteria (1.88%), Staphylococcus (0.58%) and Prevotellaceae (0.06%). The dominant bacterial genera in the WEA samples were Lactobacillus (16.24%), Streptococcus (12.78%), Clos-tridiales (5.48%), Prevotellaceae (4.73%), Corynebacteriaceae (4.14%), Cyanobacteria (2.54%), Aerococcus (1.70%), Terrisporobacter (1.60%), Staphylococcus (0.72%) and Psy-chrobacter (0.08%). The dominant bacterial genera in the FAT samples were Clostridiales (29.44%), Corynebacteriaceae (9.97%), Streptococcus (9.56%), Aerococcus (5.49%), Terrisporobacter (3.40%), Lactobacillus (2.12%), Psychrobacter (1.95%), Cyanobacteria (1.87%), Streptococcus (1.74%) and Prevotellaceae (0.12%) (lines 210-220).

  • Line 215-219 these are jour genera. Cannot be pathogenic, since there are not species determined. Provide list of relatives to your sequences.

Response 8: Thank you for your comments. In this section, we want to analyze the potential pathogenic genera that suggest potential risks. Numerous previous studies have followed the same pattern (Chen et al. 2021, Tang et al. 2021, Yan et al. 2019). You have provided extraordinary suggestions for our study, and we will pay more attention to separate sequencing and sequence analysis of individual pathogenic bacteria in future studies.

  • Line 233-235 only amount of faece aerosol load of microbe?

Response 9: Thank you for your comment, and I apologize for our inappropriate description. Our statement was too absolute, and we intended to convey the possible causes of the change in the concentration of culturable bacteria. We have changed the description as you suggested, and the new description is as follows:

This may be due to the increased activity and excretion of pigs as they grow and

increased feeding density, resulting in increased concentrations of bacterial aerosols (lines 239-241).

  • Line 252 255 How.

Response 10: Thank you for your comments. We have analyzed the possible causes of this phenomenon in conjunction with previous and published studies (Kim et al. 2019, Tang et al. 2021, Gao et al. 2015). This phenomenon is interesting and will be investigated in more depth in future studies on this phenomenon.

  • Line 256-296 can you prepare analysis of how many OTUs can be shared with pigs’ gut microbiome.

Response 11: Thank you for your comments, which provided us with extraordinary ideas for our research. Many previous studies have only analyzed microbial aerosols, while existing studies have shown that bacterial aerosols in piggery mainly come from pig feces. However, the focus of our research is to analyze the differences in the characteristics of bacterial aerosols in different types of closed pig houses. Of course, as you suggested, if we can analyze the common OTUs of intestinal microorganisms and bacterial aerosols in pigs, we can further analyze the mechanism of such changes. We will further investigate the link between gut microbes and microbial aerosols in future studies.

  • Line 270-277 this cannot be you do not have rpocies、

Response 12: Thank you for your comments. We have analyzed our results with reference to previously published studies (Tang et al. 2021, Yan et al. 2019), but you have provided us with a new way of thinking for future studies.

  • Line 294-298 see comment in results. You need to prove that at least by comparing sequences.

Response 13: Thank you for your comments. The previously published studies (Tang et al. 2021, Yan et al. 2019) did not analyze specific sequences but only potential pathogenic genera, suggesting potential pathogenic risk, but you have provided a new way of thinking for our future studies.

  • Line 306 usually staphylococcus stained are not well aerosolized. Check literature.

Response 14: Thank you for your comment, and I apologize for our carelessness. By reviewing a large number of studies, we found that most Staphylococcus bacteria are nonpathogenic bacteria, and only a few can cause diseases. We have removed the relevant text about Staphylococcus as a pathogen.

  • Can you predict how high is a rise due to the air load and exposure time.

Response 15: Thank you for your comments. The purpose of our study was to compare the differences in bacterial air microorganisms in different types of pig houses. The research related to bacterial microorganisms at different heights in a single type of pig house is still to be deepened. Thank you for your extraordinary suggestions for our study. We will consider investigating this direction in depth in our future research. We have changed the description as you suggested, and the new description is as follows:

This study showed that with the increase in weekly pig age, the concentration of airborne bacteria, the species, and the diversity index of the bacterial community in pig houses also increased gradually. Pathogenic bacteria were detected in three types of pig houses: Acinetobacter, Erysipelothrix, Streptococcus, Moraxella and Aerococcus. These bacteria with potential risk to pigs and human health should be valued. The number of disinfections should be increased to 2-3 times a week or every other day. A disinfectant should be replaced after 2-3 weeks of continuous use of another disinfectant to avoid pathogenic microorganisms producing drug resistance and reduce the disinfection effect. Spray disinfection should be used to kill airborne microorganisms in piggery to make the piggery cleaner. These measures may help to optimize the piggery environment. From the point of view of personnel, farmers should wear masks, hats and overalls to combat the effects of air pollution when working on farms. In this study, we emphasize the necessity of long-term detection of microbial aerosols in pig houses and the importance of more comprehensive research and analysis of microbial aerosols in livestock and poultry houses (lines 313-326).

Round 2

Reviewer 1 Report

Dear authors

I like the changes made to the article. They definitely increased its scientific value. Please check the references. I see some mistakes there.

Author Response

Dear Reviewers:

Thank you for your approval of our manuscript, and we appreciate your suggestions and help. The problem with the references has been improved and we hope that our changes will be approved by you.
